# C7-Prenylation of Tryptophan-Containing Cyclic Dipeptides by 7-Dimethylallyl Tryptophan Synthase Significantly Increases the Anticancer and Antimicrobial Activities

**DOI:** 10.3390/molecules25163676

**Published:** 2020-08-12

**Authors:** Rui Liu, Hongchi Zhang, Weiqiang Wu, Hui Li, Zhipeng An, Feng Zhou

**Affiliations:** 1College of Life Science, Shanxi Datong University, Datong 037009, China; liurlw@163.com (R.L.); lihuihello2000@163.com (H.L.); 2Applied Biotechnology Institute, Shanxi Datong University, Datong 037009, China; 18803521020@139.com (W.W.); zhipeng.an@126.com (Z.A.); linzhoufeng@163.com (F.Z.)

**Keywords:** cyclic dipeptides, prenyltransferase, prenylated derivatives, anticancer, antibacterial, antifungal

## Abstract

Prenylated natural products have interesting pharmacological properties and prenylation reactions play crucial roles in controlling the activities of biomolecules. They are difficult to synthesize chemically, but enzymatic synthesis production is a desirable pathway. Cyclic dipeptide prenyltransferase catalyzes the regioselective Friedel–Crafts alkylation of tryptophan-containing cyclic dipeptides. This class of enzymes, which belongs to the dimethylallyl tryptophan synthase superfamily, is known to be flexible to aromatic prenyl receptors, while mostly retaining its typical regioselectivity. In this study, seven tryptophan-containing cyclic dipeptides **1a**–**7a** were converted to their C7-regularly prenylated derivatives **1b**–**7b** in the presence of dimethylallyl diphosphate (DMAPP) by using the purified 7-dimethylallyl tryptophan synthase (7-DMATS) as catalyst. The HPLC analysis of the incubation mixture and the NMR analysis of the separated products showed that the stereochemical structure of the substrate had a great influence on their acceptance by 7-DMATS. Determination of the kinetic parameters proved that *cyclo*-l-Trp–Gly (**1a**) consisting of a tryptophanyl and glycine was accepted as the best substrate with a *K*_M_ value of 169.7 μM and a turnover number of 0.1307 s^−1^. Furthermore, docking studies simulated the prenyl transfer reaction of 7-DMATS and it could be concluded that the highest affinity between 7-DMATS and **1a**. Preliminary results have been clearly shown that prenylation at C7 led to a significant increase of the anticancer and antimicrobial activities of the prenylated derivatives **1b**–**7b** in all the activity test experiment, especially the prenylated product **4b**.

## 1. Introduction

Cyclic dipeptides (CDPs) and derivatives are widely distributed in microorganisms and exhibit diverse biologic and pharmacological activities [1,2]. A further interesting class of CDPs is indole alkaloids often containing prenyl moieties before undergoing additional modifications like cyclization, oxidation or acetylation [3]. Such as the well-known cytotoxic tryprostatin A, B and fumitremorgin C are known as the potent inhibitor of the certain cancer resistance protein [4,5,6]. Due to its higher lipophilicity, the introduction of the prenyl moiety can increase biologic activity compared to its nonallylated precursor [7]. Structural analysis shows these natural products are composed of prenyl moieties from prenyl diphosphate and indole or indoline rings from tryptophan [8,9,10,11]. Fungi are the most prolific producers that biosynthesize a family of prenylated secondary metabolites with significant biologic and pharmacological activities, such as antifungal [12], antibacterial [13,14], antiviral [15], anti-inflammatory [16] and antitumor activities [17]. Among so many biologic activities, their antitumor, antibacterial and antifungal activities are the most prominent.

Consequently, synthesis of prenylated cyclic dipeptides has drawn remarkable attention and different strategies have been developed. The synthetic routes usually deal with anhydrous or anaerobic conditions, environment hazardous chemicals and high or very low temperature [18,19,20]. Meanwhile, additional steps are usually necessary for protection and deprotection of the functional groups [21]. Therefore, we need a more efficient and gentler synthetic strategy, such as enzymatic synthesis was considered the desirable pathway.

Prenyltransferases are involved in the biosynthesis of these natural products and catalyze the regiospecific, in most cases Friedel–Crafts alkylations by transferring prenyl moieties from different prenyl donors to various acceptors [22,23]. The prenyl moieties with different carbon chain lengths (C5, C10, C15 or C20 units) can be attached in the reverse or regular pattern and further modified by cyclization, oxidation and more. Therefore, prenyltransferases play an important role in the structural diversity of such products [24,25]. Investigations in the last few years have shown that prenyltransferases catalyze the alkylation of a broad spectrum of nucleophilic aromatic substrates by electrophilic allylic pyrophosphates [26]. Prenyltransferases group share meaningful sequence identities with the dimethylallyltryptophan synthase in the biosynthesis of ergot alkaloids and are therefore termed DMATS enzymes [27,28]. Most prenyltransferases of the DMATS superfamily use dimethylallyl diphosphate (DMAPP) as a donor and L-tryptophan or tryptophan-containing cyclic dipeptides as acceptors. They are involved in the biosynthesis of diverse indole alkaloids [29,30,31]. Mechanistic studies have established that the primary center of DMAPP is attacked by the electron-rich aromatic ring with a concerted displacement of pyrophosphate to form the arenium ion intermediate, which re-aromatizes by deprotonation to form the final product [32,33]. To date, about fifty prenyltransferases from fungi and bacteria belonging to the DMATS super family have been characterized biochemically [34,35]. Structurally, the natural product prenyltransferases share a common structural motif called “ABBA” fold, in which the active site is located in the center of a huge β barrel formed by 10-strand antiparallel β-strands (surrounded by α helices) for protecting the allyl carbanion from the influence of solvents, thereby to promote catalysis [36]. These prenyltransferases also accept aromatic substrates, which differ distinctly from their natural substrates, and this feature makes them a useful tool for chemoenzymatic reactions [37,38]. For example, FtmPT [39,40], AnaPT [41], FgaPT [42], 5-DMATS [43,44], 6-DMATSSv [44] and 7-DMATS [45,46,47] from this family catalyze regiospecific prenylations of L-tryptophan at C-2, C-3, C-4, C-5, C-6 and C-7 of the indole ring, respectively. The cyclic dipeptide prenyltransferases accept in turn tryptophan-containing cyclic dipeptides as natural [48] or best substrates [49,50]. Among them, 7-DMATS is the cyclic dipeptide prenyltransferase with relatively high efficiency [44], which catalyzes a prenylation at the benzene ring in the presence of its natural prenyl donor DMAPP.

In view of these findings and in continuation of our study on active indole diketopiperazine, we reported the enzymatic synthesis of C7-prenylation tryptophan-containing cyclic dipeptides, in which isopentenyl was linked to the indole ring of different cyclic dipeptides to enhance their biologic activity. The anticancer, antibacterial and antifungal activity of the C7-prenylation tryptophan-containing cyclic dipeptides have not yet been reported and could be interesting candidates for further biologic and pharmacological investigations.

## 2. Results and Discussion

### 2.1. Prenylation of Tryptophan-Containing Cyclic Dipeptides by 7-DMATS

After 7-DMATS enzyme-catalyzed reaction, seven C7-prenylation indole diketopiperazines were synthesized (Figure 1). Through HPLC analysis, it could be found that 7-DMATS had a certain selectivity to the substrate (Table 1). *Cyclo*-l-Trp-Gly (**1a**) was the substrate with the highest conversion rate (33.6%), followed by *cyclo*-l-Trp-l-Leu (**3a**), *cyclo*-l-Trp-l-Trp (**6a**), *cyclo*-l-Trp-l-Tyr (**5a**) and *cyclo*-l-Trp-l-Pro (**7a**) as substrates, with the conversion rate of 30.2%, 28.5%, 28.1% and 25.4% and *cyclo*-l-Trp-l-Phe (**4a**) as substrate, with the lowest conversion rate (11.8%).

To prove their structures, seven enzyme products (**1b**–**7b**) were isolated on HPLC (Figure 1) and subjected to ESI-MS and NMR analyses. From the ESI-MS data, it could be seen that the molecular weights of all separated products are 68 Daltons larger than the molecular weights of their respective substrates. It proved that the structure of the synthesized product may have a mono prenylation. When the synthetic products (**1b**–**7b**) were separated from all the enzyme reactions, HPLC analysis showed that their retention time on the RP column was longer than the given substrates. Inspection of their ^1^H-NMR spectra revealed clearly the presence of signals for a regular prenyl moiety each at δ_H_ 3.49–3.54 (d, H-1′), 5.30–5.43 (t, H-2′), 1.66–1.81 (s, 3H, H-4′) and 1.56–1.75 ppm (s, 3H, H-5′). The chemical shift of H-1′ (δ_H_ 3.49–3.54) proved that it was connected to the aromatic C atom [7,25,43,46]. A detailed comparison between the **1b**–**7b** NMR spectrum and their substrates NMR spectrum showed that the doublet of H-7 has disappeared. This argument could also be deduced from the disappearance of an aromatic proton signal at H-7, which was derived from the number and coupling mode of the aromatic proton. The chemical shifts of the remaining three coupled protons on the prenylated ring was significantly different from those of C4-prenylated, but consistent with those of C7-prenylated derivatives [46,51].

Their NMR data and MS data are given as follows:

*Cyclo*-l-7-dimethylallyl-Trp-Gly (**1b**) (500 MHz, CD_3_OD, *δ*, ppm, J/Hz) 7.07 (s, H-2), 7.42 (d, *J* = 7.9, H-4), 6.93 (t, *J* = 7.5, H-5), 6.85 (dd, *J* = 7.1, 0.6, H-6), 3.44 (dd, *J* = 14.7, 3.8, H-10), 3.13 (dd, *J* = 14.7, 4.5, H-10), 4.26 (t, *J* = 4.2, 1.0, H-11), 4.12 (1H, d, *J* = 15.0, H-14a), 3.57 (1H, d, *J* = 15.0, H-14b), 3.52 (d, 7.3, H-1′), 5.42 (t,7.2, 1.4, H-2′), 1.75 (s, H-4′), 1.74 (s, H-5′); ESI-MS *m*/*z* 312.2 [M + H]^+^.

*Cyclo*-l-7-dimethylallyl-Trp-l-Ala (**2b**) (500 MHz, CD_3_OD, *δ*, ppm, J/Hz) 7.07 (s, H-2), 7.43 (d, *J* = 7.9, H-4), 6.93 (t, *J* = 7.5, H-5), 6.86 (dd, *J* = 7.1, 0.6, H-6), 3.44 (dd, *J* = 14.7, 3.8, H-10), 3.13 (dd, *J* = 14.7, 4.5, H-10), 4.26 (t, *J* = 4.2, 1.0, H-11), 3.69 (dd, *J* = 7.1, 1.5, H-14), 0.34 (d, *J* = 7.0, H-17), 3.52 (d, *J* = 7.2, H-1′), 5.39 (t, *J* = 7.2, 1.4, H-2′), 1.73 (s, H-4′), 1.73 (s, H-5′); ESI-MS *m*/*z* 326.2 [M + H]^+^.

*Cyclo*-l-7-dimethylallyl-Trp-l-Leu (**3b**) (500 MHz, CD_3_OD, *δ*, ppm, J/Hz) 7.06 (s, H-2), 7.43 (d, *J* = 8.0, H-4), 6.93 (t, *J* = 7.5, H-5), 6.87 (dd, *J* = 7.2, 0.9, H-6), 3.46 (dd, *J* = 14.7, 3.6, H-10), 3.11 (dd, *J* = 14.7, 4.6, H-10), 4.26 (ddd, *J* = 4.4, 3.4, 0.9, H-11), 3.60 (m, H-14), 1.13 (m, H-17a), 0.64 (m, H-17b), 1.75 (dd, *J* = 7.5, 1.0, H-18), 0.59 (d, *J* = 6.6, H-19), 0.44 (d, *J* = 6.6, H-20), 3.52 (d, *J* = 7.3, H-1′), 5.43 (t, *J* = 7.2, 1.4, H-2′), 1.76 (s, H-4′), 1.75 (s, H-5′); ESI-MS *m*/*z* 368.5 [M + H]^+^.

*Cyclo*-l-7-dimethylallyl-Trp-l-Phe (**4b**) (500 MHz, CDCl_3_, *δ*, ppm, J/Hz) 8.16 (s, NH-1), 6.98 (s, H-2), 7.48 (d, *J* = 8.4, H-4), 7.13 (d, *J* = 7.5, H-5), 7.05 (d, *J* = 6.9, H-6), 3.28 (dd, *J* = 14.6, 3.1, H-10), 2.62 (dd, *J* = 14.6, 8.3, H-10), 4.22 (m, H-11), 4.09 (m, H-14), 5.78 (s, NH-12), 5.72 (s, NH-15), 3.07 (dd, *J* = 13.5, 3.1, H-17a), 2.14 (dd, *J* = 13.5, 9.1, H-17b), 6.88(d, *J* = 7.0, H-19), 7.30 (m, H-20), 7.26 (m, H-21), 7.29 (m, H-22), 6.89 (d, *J* = 7.0, H-23), 3.54 (d, *J* = 7.0, H-1′), 5.30 (t, *J* = 7.2, H-2′), 1.79 (s, H-4′), 1.71 (s, H-5′); ESI-MS *m*/*z* 402.5 [M + H]^+^.

*Cyclo*-l-7-dimethylallyl-Trp-l-Tyr (**5b**) (500 MHz, CD_3_OD, *δ*, ppm, J/Hz) 7.06 (s, H-2), 7.45 (d, *J* = 7.7, H-4), 7.02 (t, *J* = 7.6, H-5), 6.93 (d, *J* = 7.0, H-6), 3.05 (dd, *J* = 14.6, 4.2, H-10), 2.93 (dd, *J* = 14.6, 5.2, H-10), 4.18 (t, *J* = 4.8, H-11), 3.81 (dd, *J* = 9.2, 3.5, H-14), 2.53 (dd, *J* = 13.6, 3.6, H-17), 1.26 (dd, *J* = 13.6, 9.2, H-17), 6.59 (d, *J* = 8.6, H-19), 6.38 (d, *J* = 8.6, H-20), 6.38 (d, *J* = 8.6, H-22), 6.59 (d, *J* = 8.6, H-23), 3.49 (d, *J* = 7.2, H-1′), 5.30 (t, *J* = 7.2, 1.4, H-2′), 1.66 (s, H-4′), 1.56 (s, H-5′); ESI-MS *m*/*z* 418.2 [M + H]^+^.

*Cyclo*-l-7-dimethylallyl-Trp-l-Trp (**6b**) (500 MHz, CDCl_3_, *δ*, ppm, J/Hz) 8.04 (s, NH-1), 6.56 (d, *J* = 2.2, H-2), 7.42(d, *J* = 7.9, H-4), 7.06 (t, *J* = 7.5, H-5), 7.01 (d, *J* = 7.1, H-6), 3.24 (dd, *J* = 14.6, 3.2, H-10), 2.41 (dd, *J* = 14.6, 8.6, H-10), 4.20 (m, H-11), 5.70 (s, NH-12), 4.19 (m, H-14), 5.75 (s, NH-15), 3.24 (dd, *J* = 14.6, 3.2, H-17), 2.51 (dd, *J* = 14.6, 8.2, H-17), 6.60 (d, *J* = 2.2, H-19), 8.09 (s, NH-20), 7.35 (d, *J* = 8.2, H-22), 7.22 (t, *J* = 7.5, H-23), 7.16 (t, *J* = 7.6, H-24), 7.58 (d, *J* = 8.0, H-25), 3.52 (d, *J* = 7.2, H-1′), 5.30 (t, *J* = 7.2, H-2′), 1.81 (s, H-4′), 1.72 (s, H-5′); ESI-MS *m*/*z* 441.2 [M + H]^+^.

*Cyclo*-l-7-dimethylallyl-Trp-l-Pro (**7b**) (500 MHz, CDCl_3_, *δ*, ppm, J/Hz) 8.15 (s, NH-1), 7.01 (s, H-2), 7.49 (d, *J* = 8.4, H-4), 7.12 (d, *J* = 7.5, H-5), 7.05 (d, *J* = 7.0, H-6), 3.27 (dd, *J* = 14.8, 3.2, H-10), 2.61 (dd, *J* = 14.8, 8.2, H-10), 4.22 (m, H-11), 5.78 (s, NH-12), 4.05 (1H, d, *J* = 8.2, H-14), 5.71 (s, NH-15), 1.99 (2H, m, H-17), 2.18 (2H, m, H-18), 3.35 (2H, m, H-19), 3.54 (d, *J* = 6.9, H-1′), 5.30 (t, *J* = 7.2, H-2′), 1.79 (s, H-4′), 1.71 (s, H-5′); ESI-MS *m*/*z* 352.2 [M + H]^+^.

In addition, the ^1^H-NMR data of **1b** and **2b** were consistent well with those of isopentenyl tryptophan derivatives synthesized by Fan et al. [52]. The ^1^H-NMR data of **3b**, **5b** and **7b** were similar to those of the enzyme products of CTrpPT [45]. The ^1^H-NMR data of **4b** and **6b** in CDCl_3_ corresponded perfectly to those of the enzyme products of CTrpPT, respectively, whose structures have been elucidated by NMR analysis including HSQC and HMBC in that study [46]. The comparison results of NMR data were very consistent, which further verified the correct structure of the enzyme product. These compounds were therewith identified as C7-prenylated derivatives (Figure 1).

### 2.2. Kinetic Parameters of 7-DMATS

To find out suitable conditions for determination of kinetic parameters, dependency of product formation on incubation time was demonstrated with 4.0 μM 7-DMATS in the presence of 1 mM **1a** and 2 mM DMAPP. Linear dependency of up to 120 min was observed in this experiment. Michaelis–Menten kinetics parameters for DMAPP and **1a**–**7a** are shown in Table 2. For seven selected cyclic dipeptides (**1a**–**7a**), kinetic parameters including Michaelis–Menten constants (*K*_M_) and turnover numbers (*k_c_*_at_) were determined and calculated from Lineweaver–Burk, Hanes–Woolf and Eadie–Hofstee plots (Table 2; Figure 2; Appendix A).

The reactions catalyzed by 7-DMATS apparently followed Michaelis–Menten kinetics. 7-DMATS showed a high affinity to its prenyl donor DMAPP with a *K*_M_ value of 79.6 μM and a *k_c_*_at_ of 0.0693 s^−1^.

Simultaneously, the *K*_M_ value of DMAPP was almost stable in all enzyme synthesis reactions of substrates. This result also proved that DMAPP acted as a stable isopentenyl donor in the prenylation reaction [32,53]. The order of *K*_M_, *k_c_*_at_ and *k_c_*_at_/*K*_M_ for seven substrates was inconsistent. In the cases of the tested aromatic substrates, 7-DMATS showed the highest affinity to **1a** with a *K*_M_ value of 169.7 μM, a turnover number *k_c_*_at_ of 0.1307 s^−1^ and *k_c_*_at_/*K*_M_ ratio 770.1 s^−1^M^−1^. It is worth noting that **4a** had the lowest *K*_M_ value, but its *k_c_*_at_ value and *k_c_*_at_/*K*_M_ ratio were lower than **1a**. Comprehensive analysis displayed that 7-DMATS showed the average affinity to **1a**. *Cyclo*-l-Trp-l-Ala (**2a**) was the substrate having the lowest affinity to 7-DMATS according to the *k_c_*_at_ (0.0127 s^−1^), *K*_M_ (867.8 μM) and *k_c_*_at_/*K*_M_ (14.6 s^−1^M^−1^). For the other tryptophan-containing cyclic dipeptides (**3a**, **5a–6a**), *K*_M_ values between 225.8 and 823.2 μM and turnover numbers in the range of 0.0413 and 0.0586 s^−1^ were determined (Table 2). Two significantly higher *K*_M_ value of 867.8 and 880.1 μM was calculated for *cyclo*-l-Trp-l-Ala (**2a**) and *cyclo*-l-Trp-l-Pro (**7a**) with the relative catalytic efficiency of 1.9% and 3.01% of that of **1a**.

### 2.3. Docking with **1a**–**7a**

As shown in Table 3, the 3D simulated docking of the ligand (**1a–7a**) and the receptor protein (7-DMATS) with the lowest binding free energy were determined. The seven substrates were well bound to the protein receptor and the H atom of the indole ring of tryptophan was the common active site, which formed hydrogen bonds with amino acid residues such as GLU89, PRO313 and LEU81 of the protein receptor. The molecular mechanics Poisson–Boltzmann surface area (MM-PBSA) was commonly used to calculate binding free energy (ΔG_bind_) of docking with the receptor and ligand molecules. The calculation of combined free energy was based on four components, namely van der Waals force contribution (ΔG_vdw_), electrostatic contribution (ΔG_ele_), desolvated polar part (ΔG_polar_) and nonpolar contribution (ΔG_nonpolar_) [54]. The lower ΔG_bind_ value showed that the affinity between the receptor and ligand was highest [55].

The lowest **ΔG_bind_** of the docking complex was calculated by LeDock and the hydrophobic amino acid residues of **1a–7a** and 7-DMATS was selected by PyMOL 1.5 as shown in Figure 3. Following the principle of the lowest ΔG_bind_, combined with the number and type of bonds, it can be concluded that the catalytic effect of 7-DMATS on the target substrate was **1a** > **4a** > **6a** > **5a** > **3a** > **7a** > **2a**. For Kinetic parameters analysis, the order of *k*_cat_/*K*_M_ ratio was **1a** (100%) > **4a** (40.77%) > **6a** (23.75%) > **5a** (10.7%) > **3a** (9.25%) > **7a** (3.01%) > **2a** (1.90%). The agreement between the experimental data and the analysis of the molecular docking model also confirmed the accuracy of this docking model and provided a certain predictive model for the further 7-DMATS enzymatic reaction.

### 2.4. Anticancer Activity

Anticancer activity in vitro of the substrates **1a**–**7a** and prenylated substances **1b**–**7b** against four cancer cell lines was determined by MTT assay after 72 h treatment as shown in Table 4. Whereas most of the non-prenylated compounds showed IC_50_ values > 200 μM, all of the tested prenylated enzyme products were more toxic towards the four cancer cell lines with IC_50_ values in the lower to medium micromolar range for MCF-7 or the higher micromolar range for HeLa, HepG2 and A549 cells. With a few exceptions, for example **1b** and **4b** in the assays with HeLa and A549, all prenylated substances showed similar toxicity for the given cell lines. Among them, Human breast cancer cells MCF-7 had a higher sensitivity for prenylated substances than human cervical cell lines HeLa with IC_50_ values from 32.3 to 45.6 μM (except **7b**). Compound **4b** recorded highest activity against all the test cancer cell lines, especially for A549 and MCF-7. Its IC_50_ value is half lower than other prenylated substances.

The inhibition ratios of these compounds at different concentrations to the proliferation of human cancer cells HeLa, HepG2, A549, and MCF-7 were evaluated in Figure 4. Compound **4b** exhibited the most significant anticancer activity against the four cancer cells. At 100 μM, the inhibition rates of **4b** for HeLa, HepG2, A549, and MCF-7 cells were 73.77%, 72.93%, 86.23% and 99.18%, respectively. Moreover, **1b** and **7b** showed better anticancer activities against HepG2 cells with inhibition ratios reaching more than 65.05% and 70.0% at 100 μM. In particular, the inhibition ratios of **1b**–**7b** against MCF-7 exceeded 90% at 100 μM. Combined with the structural features of the prenylated substances, the presence of isopentenyl at the C-7 position of these substances was critical to the cytotoxicity activities.

### 2.5. Antibacterial Activity

The substrates and prenylated substances were tested for antibacterial activity against Gram-positive bacteria and Gram-negative bacteria using standard methods. The minimal inhibitory concentration (MIC) values of all the compounds were presented in Table 5 and Table 6. All the prepared prenylated substances showed relatively higher antibacterial activities than their substrates. The activity of the most substrates was comparable to the standard antibiotic ampicillin and much lower than ciprofloxacin. However, the activity of the most tested prenylated substances was much higher than the standard antibiotics ampicillin and the equivalent of ciprofloxacin.

Gram-positive strains *B. subtilis*, *S. aureus* and *S. simulans* showed relatively high sensitivities toward the synthesized prenylated compounds, with MIC values from 0.5 μg·mL^−1^ to 64 μg·mL^−1^. Furthermore, 2b and 4b showed prominent activities with MIC values of 0.5 μg·mL^−1^ against *B. subtilis* (**4b**), *E. coli* (**2b**) and *P. mirabilis* (**2b**), 1 μg·mL^−1^ against *S. epidermis* (**4b**), *E. coli* (**4b**) and 2 μg·mL^−1^ against *S. aureus* (**4b**), *S. simulans* (**4b**), *K. pneumoniae* (**2b**) and *P. aeruginosa* (**4b**). This activity value of **2b** and **4b** was significantly better than ciprofloxacin. Derivative **1b** was active against all the test bacteria and best activity of this compound was recorded against *K. pneumoniae* (1 μg·mL^−1^), followed by *E. coli* and *P. mirabilis* (2 μg·mL^−1^). Derivative **2b** presented highest activity against *E. coli* and *Proteus mirabilis* (0.5 μg·mL^−1^). Derivatives **3b** and **7b** are active only against seven test bacteria and highest activity was recorded against *S. simulans* (32 μg·mL^−1^) and *S. aureus* (2 μg·mL^−1^), respectively. Derivative **4b** presented highest activity against *B. subtilis* (0.5 μg·mL^−1^). Derivative **5b** was active against all the test bacteria with MIC values from 1 μg·mL^−1^ to 16 μg·mL^−1^ and best activity was recorded against *P. aeruginosa* (1 μg·mL^−1^), followed by *S. epidermis* and *E. coli* (2 μg·mL^−1^). Derivative **6b** was active only against six test bacteria and highest activity was recorded against *S. aureus* (1 μg·mL^−1^).

### 2.6. Antifungal Activity

Antifungal activity of the substrates and prenylated substances against eight fungi and corresponding MIC values were indicated in Table 7 and Table 8. It appeared that all the prepared prenylated substances showed relatively higher antifungal activities than the substrates. The activity against medically important fungi of the most substrates was much lower than standard fungicide Amphotericin B. Moreover, the activity of the most tested prenylated substances was much higher and the equivalent of Amphotericin B. The results of activity against agricultural fungi were also similar to medical fungi and the most tested prenylated substances recorded higher antifungal activity than the standard fungicide Bavistin against certain fungi. In particular, **2b** and **4b** showed extremely significant antifungal activity.

The microorganism that presented highest sensitivity toward 1b was *R. solani* and *P. expansum* with MIC values 2 μg·mL^−1^. Derivative **2b** exhibited significant activity against all fungi with MIC values from 0.5 μg·mL^−1^ to 4 μg·mL^−1^, against *R. solani* (0.5 μg·mL^−1^), *P. expansum* (0.5 μg·mL^−1^), *C. albicans* (1 μg·mL^−1^) and *A. Brassicae* (1 μg·mL^−1^). Derivative 3b exhibited best MIC values against *A. flavus* (2 μg·mL^−1^) and *P. expansum* (2 μg·mL^−1^). Interestingly, **4b** recorded significantly higher antifungal activity against test pathogens in impressive low concentration and best activity was *T. rubrum* and *F. oxysporum* (0.5 μg·mL^−1^), followed by *C. gastricus*, *R. solani* and *P. Expansum* (1 μg·mL^−1^), after that, *A. brassicae* (2 μg·mL^−1^). Above activity values of **2b** and **4b** were significantly better than Amphotericin B and Bavistin. Derivative **5b** was active against seven test fungi and highest activity was recorded against *F. oxysporum* and *P. expansum* (16 μg·mL^−1^). Derivative **6b** was also active against seven test fungi and best activity was *C. albicans* and *R. Solani* (4 μg·mL^−1^). Derivative **7b** was active against all the test fungi with MIC value of 1 μg·mL^−1^ against *C. gastricus*, followed by 2 μg·mL^−1^ against *F. oxysporum* and *P. Expansum* of 0.5 μg·mL^−1^.

It was clearly that prenylation at the indole ring C-7 led to a significant increase in anticancer, antibacterial and antifungal activities. In addition to L-tryptophan, most of these substances (**1a**–**7a**) also contain a second amino acid and form a cyclic dipeptide with the structure of 2,5-diketopiperazine (2,5-DKPs). The 2,5-DKPs are a natural privileged structure with the ability to bind to multiple receptors. These small, conformationally rigid chiral templates have multiple H-bond acceptor and donor functional groups and have multiple sites for structural modification of various functional groups that define stereochemistry [1]. These characteristics enable them to bind to a variety of receptors with high affinity and display a wide range of biologic activities.

It has been hypothesized speculated that due to the increase in the hydrophobicity of 2,5-DKPs by adding isoprene groups, the prenylated molecules are better distributed on the membrane than the non-prenylated molecules and are more closely related to the target protein [8,9,35,53]. In the structures (**1b**–**7b**), the prenyl moieties were connected via its C1 to indole ring C7. Prenylation not only improved the affinity for biomembranes, but also enhanced the interaction with proteins of these substances (**1a**–**7a**), and therefore C7-prenylation of tryptophan-containing cyclic dipeptides significantly increased the biologic activity.

### 2.7. Optimization of Enzyme-Catalyzed Reaction Conditions

Biologic activity results showed that *cyclo*-l-7-dimethylallyl-Trp-l-Phe (**4b**) had the best cytotoxicity, antibacterial and antifungal activities. Hence, the enzyme-catalyzed synthesis conditions of **4b** were optimized (Figure 5; Appendix A). 

Thus, the effects of pH and temperature on prenylation activity of 7-DMATS were measured using **4a** as acceptor and DMAPP as prenyl donor. First, in order to accurately find the influencing factors of the enzyme reaction, the enzymatic reaction time was determined to be 18 h for further investigation of the biochemical properties in these assays (Figure 5A). The relative activities of 7-DMATS were then determined at different pH, and the optimum pH value was between 8.0 and 9.0 (Tris-HCl buffer). The optimum temperature for 7-DMATS was 35 °C and its activity decreased rapidly at above 37 °C. It is worth noting that at different temperatures and pH values, the enzyme activities show great differences. In other words, enzyme activity of 7-DMATS was strongly affected by pH and temperature, and the result was consistent with previous literatures [56,57]. In addition, no 7-DMATS activity was observed without the addition of divalent cations, demonstrating the enzymatic dependence on divalent metal ions for the activity of 7-DMATS. Subsequently, different divalent cations were tested and 7-DMATS activity was observed to decrease in the order of Mg^2+^ > Ca^2+^ > Mn^2+^ > Fe^2+^ > Ba^2+^ > Zn^2+^, and no product was detected with the presence of Cu^2+^. All these results were similar to the previously determined plant isopentenyl transferase activity [58,59,60].

## 3. Materials and Methods

### 3.1. General

DMAPP was prepared according to the method described for geranyl diphosphate [61]. *Cyclo*-l-Trp-l-Tyr was synthesized according to a protocol described previously [62]. Other cyclic dipeptides were prepared as described elsewhere [63] or purchased from Bachem (Bubendorf, Switzerland). All the chemicals used for extraction, high performance liquid chromatography (HPLC) were purchased from Merck Limited, Germany. All other reagents and chemicals used in this study were the highest purity. The standard antibiotics ciprofloxacin, ampicillin and amphotericin B and MTT were obtained from Amersco, Inc. (Solon, OH, USA). Yeast extract and tryptone were from Oxoid, Ltd. (Basingstoke, Hampshire, UK). RPMI-1640 medium, and fetal bovine serum (FBS) were from Gibco Invitrogen Corporation (Carlsbad, CA, USA). Deionized water (Milli-Q, Millipore, Bedford, MA, USA) was used to prepare aqueous solutions. MIC was detected using a BIO-RAD 680-Microplate reader (Beijing Yuanye Bio. Co., Ltd., Beijing, China).

### 3.2. Pathogen Microbial and Cell Lines

All the test microorganisms were purchased from American Type Culture Collection, Virginia, American and details as follows, Gram-positive bacteria: *Bacillus subtilis* ATCC 23857, *Staphylococcus aureus* ATCC 12600, *Staphylococcus epidermis* ATCC 51,625 and *Staphylococcus simulans* ATCC 27,848; Gram-negative bacteria: *Escherichia*
*coli* ATCC 35218, *Klebsiella pneumoniae* ATCC 43816, *Proteus mirabilis* ATCC 21100, *Pseudomonas aeruginosa* ATCC 10,145; Medically important fungi: *Aspergillus flavus* ATCC 204304, *Candida albicans* ATCC 10231, *Cryptococcus gastricus* ATCC 32,042 and *Trichophyton rubrum* ATCC 28,191; Agriculturally important fungi: *Fusarium oxysporum* ATCC 14838, *Rhizoctonia solani* ATCC 10,182 and *Penicillium expansum* ATCC 16104, *Alternaria brassicae* ATCC 66,981. The test bacteria were maintained on nutrient agar slants and test fungi were maintained on potato dextrose agar slants.

HeLa and HepG2 cells were provided by the Cell Center of the Fourth Military Medical University (Xi’an, China). A549 and MCF-7 cells were from the Chinese Academy of Sciences (Shanghai, China). The test cell lines were maintained on RPMI-1640 medium.

### 3.3. Overproduction and Purification of 7-DMATS as Well as Conditions for Enzymatic Reactions

pGEM-T and pQE60 were obtained from Promega and Qiagen, respectively. A Uni-ZAP XR premade library of *Aspergillus fumigatus* strain B5233 (ATCC 13073) was purchased from Stratagene and used to obtain phagemids as cDNA templates for PCR amplification. *Escherichia coli* XL1 Blue MRF9 (Stratagene) was used for cloning and expression experiments, and it was grown in liquid Luria–Bertani (LB) medium with 1.5% (*w*/*v*) agar, at 37 °C [64]. The 7-DMATS was overproduced in *E. coli* and purified as described by Kremer et al. [64]. Standard enzyme assays were carried out in the reaction mixture (50 μL) containing 50-mM Tris-HCl, pH 7.5, 5-mM CaCl_2_ or MgCl_2_, 1-mM DMAPP, 1-mM substrates and 5.0 μg (950 nM) purified recombinant 7-DMATS. After incubation for 16 h at 37 °C, the reaction was quenched by the addition of trichloroacetic acid (1.5 M, 5 μL). After removal of the protein by centrifugation (15,000 g, 15 min, 4 °C), the enzymatic products were analyzed by HPLC. The assays for isolation of enzymatic products were carried out in 3-mL reaction mixtures containing 50-mM Tris-HCl, pH 7.5, 5-mM CaCl_2_, 1-mM DMAPP, 1-mM substrates and 150 μg 7-DMATS. After incubation for 16 h at 37 °C, the reactions were quenched by the addition of 100 μL trichloroacetic acid (1.5 M) and centrifuged (15,000 g, 10 min, 4 °C) for the removal of the protein.

### 3.4. HPLC Conditions for Analysis and Isolation of the Enzyme Products

Using a Multospher 120 RP-18 column (120 × 4 mm, 5 μm), the enzyme product of the incubation mixture was analyzed by HPLC on an Agilent 1200 at a flow rate of 1 mL·min^−1^. The mobile phase consisted of Water as solution A and methanol as solution B. To analyze the enzyme product, used solvent B with a linear gradient of 50%–80% (*v/v*) within 10 min and then used solvent B with a linear gradient of 80%–100% (*v/v*) within 5 min. Then the column was washed with 100% solvent B for 5 min and equilibrated with 50% (*v/v*) solvent B for 5 min. The conversion rate of the enzyme reaction was calculated by the ratio of the peak areas of the product to the sum and substrate detected at 277 nm. The enzymatic products were separated by HPLC (COSMOSIL 5C18 MS-II reverse phase column, 250 × 10 mm, 3.0 mL·min^−1^, 277 nm). Gradient elution was performed from 50% to 100% solvent B in 10 min with flow rate 2.5 mL·min^−1^. After washing with 100% solvent B for 10 min, equilibrate the column with 50% solvent B for 10 min with flow rate 2.5 mL·min^−1^. The methanol and trifluoroacetic acid were removed in vacuo, and the water was removed by lyophilization to obtain the products.

### 3.5. Spectroscopic Analysis

ESI mass spectra were obtained using a Thermo Scientific LCQ FLEET mass spectrometer equipped with an electrospray ion source and controlled by Xcalibur software (Thermo Fisher Scientific, Waltham, MA, USA). Proton nuclear magnetic resonance spectra (^1^H-NMR) was obtained using a Bruker Avance DMX 500 MHz/125 MHz spectrometer (Bruker, Billerica, MA, USA).

### 3.6. Determination of the Kinetic Parameters

The assays for determination of the kinetic parameters of *cyclo*-l-Trp-Gly (**1a**), *cyclo*-l-Trp-l-Ala (**2a**), *cyclo*-l-Trp-l-Leu (**3a**), *cyclo*-l-Trp-l-Phe (**4a**), *cyclo*-l-Trp-l-Tyr (**5a**), *cyclo*-l-Trp-l-Trp (**6a**) and *cyclo*-l-Trp-l-Pro (**7a**), contained 5 mM CaCl_2_, 2 mM DMAPP, 2.0 μM (**1a**, **2a**, **6a**) or 4.0 μM (**3a**–**5a**, **7a**) 7-DMATS and aromatic substrates at final concentrations of 0.01, 0.025, 0.05, 0.1, 0.25, 0.5, 1, 2.5 and 5.0 mM. For determination of the kinetic parameters of DMAPP, 2.0 μM 7-DMATS, 1 mM **1a**, 5 mM CaCl_2_ and DMAPP at final concentrations of up to 5.0 mM were used. Incubations were carried out at 37 °C for 60 min (**1a**, **6a**) or 120 min (other substrates). All experiments were performed in triplicate. Apparent Michaelis–Menten constants (*K*_M_), turnover numbers (*k*_cat_) and *k*_cat_/*K_M_* values were calculated from Lineweaver–Burk, Hanes–Woolf and Eadie–Hofstee plots [65] by using OriginPro 2017 software. The term *k_c_*_at_/*K*_M_ was used as a specificity constant to compare the relative reaction rates of different substrates [66].

### 3.7. Molecular Docking

A homology model of 7-DMATS was built using the experimental structure of the FgaPT2 complex with the substrate tryptophan (PDB ID: 3I4X) [67] as a homologous template by the Molecular Operating Environment (MOE). To consider the hydrogen bond network in the substrate binding pocket, both substrate molecules were also included during model building [68]. Next, the prenylated pyrophosphate analog as a co-substrate mimicry in the model structure was replaced with pyrophosphate as a coproduct molecule to elucidate the binding mode of tryptophan-containing cyclic dipeptides by docking calculations. The software LeDock (http://www.lephar.com) was used for the docking studies due to its high speed and accuracy [69]. During optimization, the product molecule and the side chain atoms around the binding site were treated as flexible. The binding pose with the lowest docking energy was adopted as a predicted binding mode. The docking results were analyzed and visualized using PyMOL 1.5 (http://www.pymol.org).

### 3.8. Anticancer Assay

The MTT (3-(4, 5-dimethyl thiazol-2-yl)-2, 5-diphenyl tetrazolium bromide) assay [70] with slight modification was used to determine the inhibition effects of substrates **1a**–**7a** and prenylated substances **1b**–**7b**. HeLa, HepG2, A549 and MCF-7 were used for testing. Briefly, cells (4.0 × 10^3^ cells per well) were seeded in 150 μL of the RPMI-1640 medium in 96-well plates, treated with drugs for 72 h and after incubation, cytotoxicity was measured. For this after removing the drug containing media, 20 μL of MTT solution (5 mg·mL^−1^ in PBS) and 75 μL of complete medium were added to wells and incubated for 4 h under similar conditions. At the end of incubation MTT lysis buffer was added to the wells (0.1 mL well^−1^) and incubated for another 4 h at 37 °C. At the end of incubation, the optical densities at 570 nm were measured using a plate reader (model 680, BIO-RAD, Hercules, CA, USA). The relative cell viability in percentage was calculated according to the formula below: [(*Acontrol* − *Atest*)/*Acontrol*] × 100%(1) where *Acontrol* and *Atest* are the optical densities of the control and the test groups, respectively. All assays were done in triplicate.

### 3.9. Antibacterial Assay

The bacterial strains were kept in liquid nitrogen (−196 °C) in a Luria-Broth (LB) medium (5 g/L yeast extract, 10 g/L bactopeptone and 10-g/L sodium chloride) containing 15% glycerol. Prior to the experiment, the bacterial strains were grown on LB agar plates at 37 °C. The minimum inhibitory concentration of the synthesized compounds was determined according to the method described by the Clinical and Laboratory Standards Institute [71], with some modifications. Two-fold serial dilutions of the antibiotics and peptide compounds were made with LB medium to give concentrations ranging from 0.5 to 1024 μg·mL^−1^. Hundred microliters of test bacterial suspension were inoculated in each tube to give a final concentration of 1.5 × 10^6^ CFU·mL^−1^. The growth was observed both visually and by measuring OD at 630 nm after 24 h incubation at 37 °C. The lowest con-centration of the test compound showing no visible growth was recorded as the MIC. Triplicate sets of tubes were maintained foreach concentration of the test sample. One well containing bacterial cells and DMSO without any test compounds (growth control), and one well containing only growth medium (sterility control), were used as controls. Ampicillin and ciprofloxacin were used as positive control. The experiments were repeated at least thrice.

### 3.10. Antifungal Assay

Similar to antibacterial activity test method. The MIC was performed by broth microdilution methods asper the guidelines of Clinical and Laboratory Standard Institute [72,73] with RPMI 1640 medium containing L-glutamine, with-out sodium bicarbonate and buffered to pH 7.0. Two-fold serial dilutions of the peptide compounds were prepared in media in amounts of 100 μL per well in 96-well microtiter plates. The test fungal suspensions were further diluted in media, and a 100 μL volume of this diluted inocula was added to each well of the plate, resulting in a final inoculum of 0.5 × 10^4^ to 2.5 × 10^4^ CFU·mL^−1^ for test fungi. The final concentration of the peptide compounds ranged from 0.5 to 1024 μg·mL^−1^. The medium without the agents was used as a growth control and the blank control used contained only the medium. Amphotericin B and Bavistin served as the standard drug controls for against medically important fungi and against agriculturally important fungi, respectively. The microtiter plates were incubated at 35 °C for 48 h for Candida species and 30 °C for 72 h for other fungi. The plates were read using ELISA, and the MIC was defined as the lowest concentration of the antifungal agents that prevented visible growth with respect to the growth control. The experiments were repeated at least thrice.

### 3.11. Optimize Enzymatic Reactions Conditions

To investigate the optimal reaction pH of recombinant 7-DMATS to **4a**, enzymatic reactions were performed in various reaction buffers ranged in pH values from 4.0–6.0 (citric acid–sodium citrate buffer), 6.0–8.0 (Na_2_HPO_4_-NaH_2_PO_4_ buffer), 7.0–9.0 (Tris-HCl buffer) and 10.0–11.0 (Na_2_CO_3_-NaHCO_3_ buffer). To optimize the reaction temperature, the enzymatic reactions were incubated at different temperatures (4–50 °C). To test the dependence of divalent ions for enzyme activity, different cations (Mg^2+^, Mn^2+^, Fe^2+^, Ba^2+^, Ca^2+^, Zn^2+^, Cu^2+^) in the final concentration of 50-mM were added, respectively. All enzymatic reactions were conducted with DMAPP as donor and compound 4a as acceptor. All experiments were performed in triplicate and the enzymatic mixtures were subjected to HPLC analysis.

## 4. Conclusions

Seven diketopiperazines of L-tryptophan series were used as substrates (**1a**–**7a**) and seven prenylated substances (**1b**–**7b**) were synthesized by 7-DMATS. Through the conversion rate and HPLC analysis, we could find that 7-DMATS had a certain selectivity to the substrate. The kinetic parameters and the molecular docking were used to analyze the reasons for the selective catalysis of 7-DMATS to the substrate, and the difference in enzyme catalytic efficiency was also verified. *Cyclo*-l-Trp–Gly (**1a**) consisting of a tryptophanyl and glycine was accepted as the best substrate with a *K_M_* value of 169.7 μM and a turnover number of 0.1307 s^−1^. Docking studies simulated the prenyl transfer reaction of 7-DMATS and it could be concluded that the highest affinity between 7-DMATS and 1a with low docking energy ΔG_bind_ (−6.31 kcal·mol^−1^). Bioactivity assays showed that the prepared prenylated substances (**1b**–**7b**) displayed relatively higher activities than substrates (**1a**–**7a**). Our results showed clearly that prenylation at the indole ring C-7 led to a significant increase in anticancer, antibacterial and antifungal activities. Among them, the activity of **4b** was determined to be the highest. The single-factor method indicated that the best enzyme catalysis conditions of **4a** were reaction time 18 h, temperature 35 °C, pH 8.0 (Tris-HCl buffer), 5 mmol·L^−1^ added of MgCl_2_. These results provided basic data for subsequent enzymatic synthesis of more prenyl indole diketopiperazine.

## Figures and Tables

**Figure 1 molecules-25-03676-f001:**
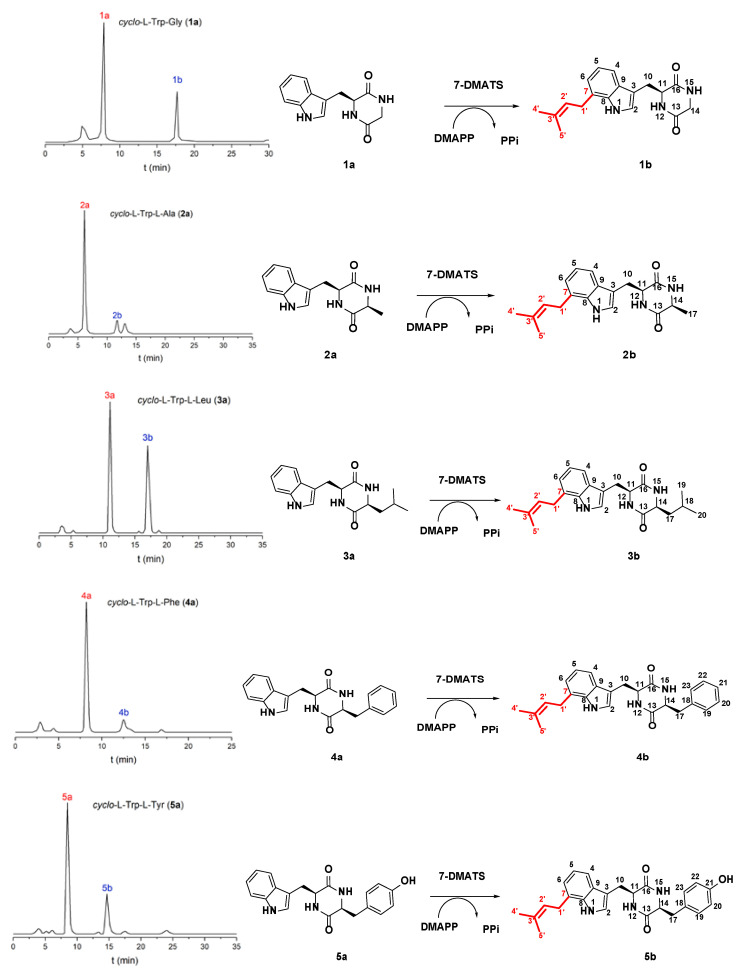
HPLC analysis of the incubation mixtures of selected substrates (**left**) and prenylation reactions catalyzed by 7-DMATS (**right**).

**Figure 2 molecules-25-03676-f002:**
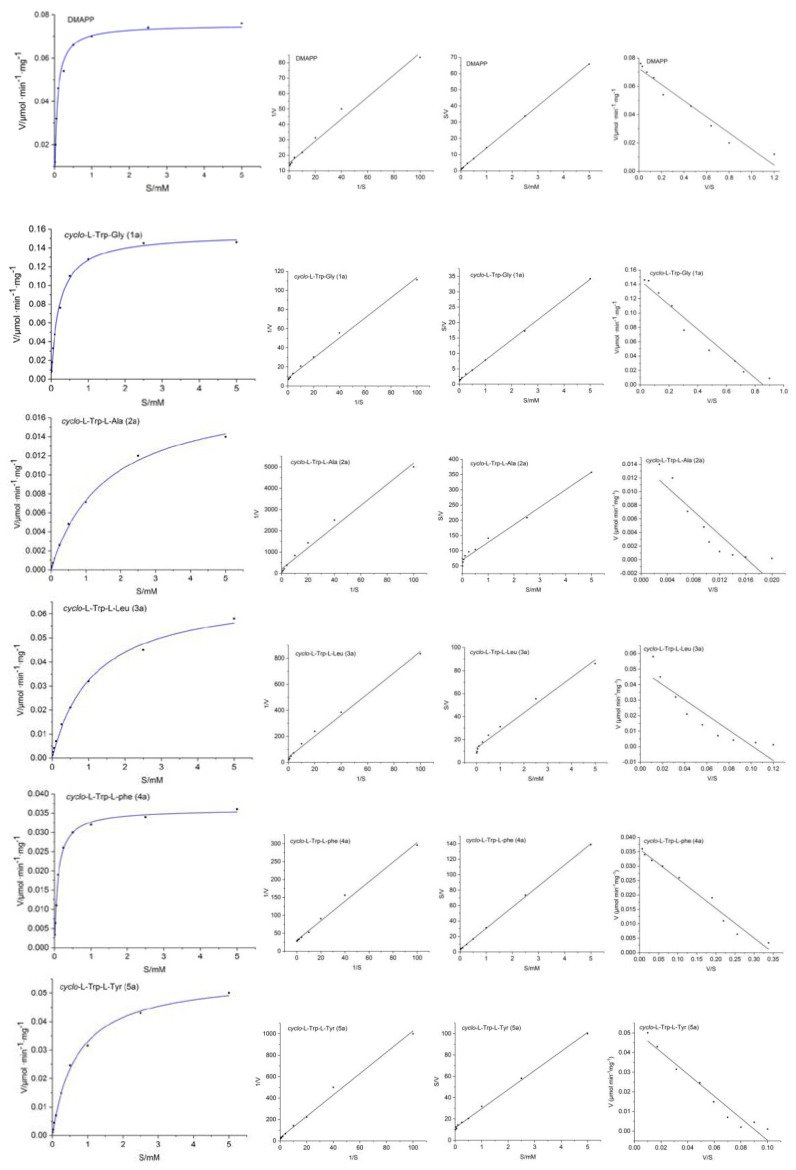
Dependency of the product formation on dimethylallyl diphosphate (DMAPP), **1a**–**7a** concentrations. Michaelis–Menten equation, Lineweaver–Burk, Hanes–Woolf and Eadie–Hofstee plots of DMAPP, **1a**–**7a** (from left to right).

**Figure 3 molecules-25-03676-f003:**
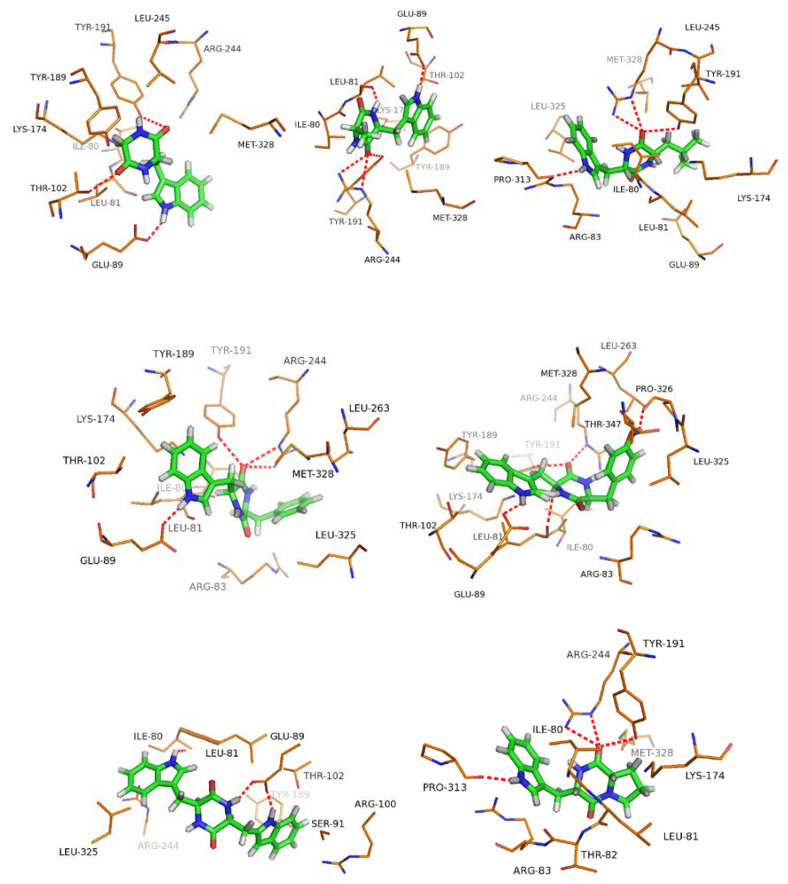
(**1a**–**7a**) Binding pattern of 7-DMATS with indole diketopiperazines. Red dotted line indicated the hydrogen bond.

**Figure 4 molecules-25-03676-f004:**
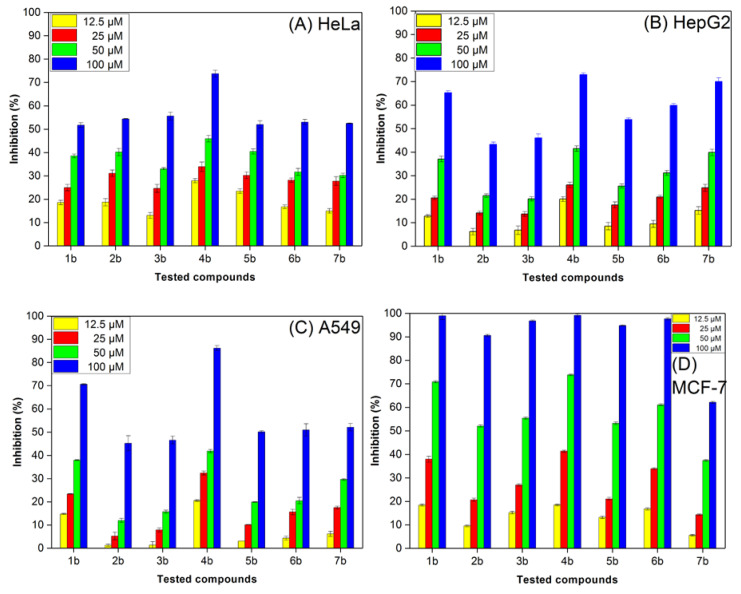
Inhibition ratios of prenylated tryptophan-containing cyclic dipeptides to the proliferation of (**A**) HeLa, (**B**) HepG2, (**C**) A549 and (**D**) MCF-7 cell lines.

**Figure 5 molecules-25-03676-f005:**
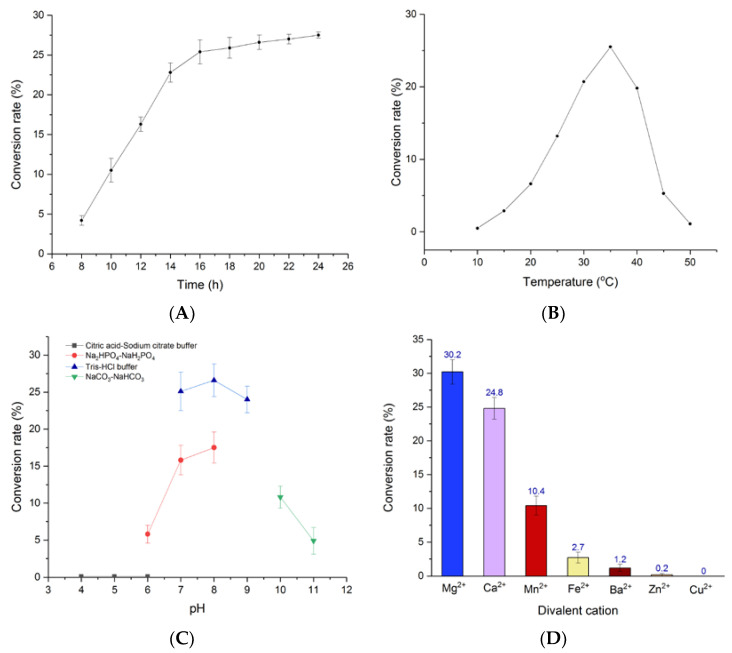
Enzyme properties of 7-DMATS. (**A**) Effects of enzymatic reaction time on 7-DMATS activity; (**B**) pH-Dependence of 7-DMATS activity; (**C**) effects of temperature on 7-DMATS activity; (**D**) divalent cation requirement on 7-DMATS activity.

**Table 1 molecules-25-03676-t001:** Product yields 7-dimethylallyl tryptophan synthase (7-DMATS) reactions.

Substrate	Product	The Final Conversion Rate (%)
*cyclo*-l-Trp-Gly (**1a**)	*cyclo*-l-7-dimethylallyl-Trp-Gly (**1b**)	33.6
*cyclo*-l-Trp-l-Ala (**2a**)	*cyclo*-l-7-dimethylallyl-Trp-l-Ala (**2b**)	20.2
*cyclo*-l-Trp-l-Leu (**3a**)	*cyclo*-l-7-dimethylallyl-Trp-l-Leu (**3b**)	30.2
*cyclo*-l-Trp-l-Phe (**4a**)	*cyclo*-l-7-dimethylallyl-Trp-l-Phe (**4b**)	11.8
*cyclo*-l-Trp-l-Tyr (**5a**)	*cyclo*-l-7-dimethylallyl-Trp-l-Tyr (**5b**)	28.1
*cyclo*-l-Trp-l-Trp (**6a**)	*cyclo*-l-7-dimethylallyl-Trp-l-Trp (**6b**)	28.5
*cyclo*-l-Trp-l-Pro (**7a**)	*cyclo*-l-7-dimethylallyl-Trp-l-Pro (**7b**)	25.4

**Table 2 molecules-25-03676-t002:** Kinetic parameters of 7-DMATS to selected substrates.

Substrate	*K*_M_ (μM)	*V*_max_ (M s^−1^)	*k*_cat_ (s^−1^)	*k*_cat_/*K*_M_ (s^−1^M^−1^)	*k*_cat_/*K*_M_ (%)
DMAPP	79.6	1.2 × 10^−9^	0.0693	870.6	113.1
*cyclo*-l-Trp-Gly (**1a**)	169.7	2.42 × 10^−9^	0.1307	770.1	100
*cyclo*-l-Trp-l-Ala (**2a**)	867.8	2.35 × 10^−10^	0.0127	14.6	1.90
*cyclo*-l-Trp-l-Leu (**3a**)	823.2	1.09 × 10^−9^	0.0586	71.2	9.25
*cyclo*-l-Trp-l-Phe (**4a**)	102.9	5.98 × 10^−10^	0.0323	314.0	40.77
*cyclo*-l-Trp-l-Tyr (**5a**)	562.4	8.58 × 10^−10^	0.0464	82.4	10.70
*cyclo*-l-Trp-l-Trp (**6a**)	225.8	7.65 × 10^−10^	0.0413	182.9	23.75
*cyclo*-l-Trp-l-Pro (**7a**)	880.1	3.80 × 10^−10^	0.0205	23.3	3.01

**Table 3 molecules-25-03676-t003:** Relations between ligands and residues of 7-DMATS.

Compound	Binding Free Energy ΔG_bind_(kcal·mol^−1^)	Hydrogen Bonding
**1a**	−6.31	TYR191, THR102, GLU89
**2a**	−4.78	ARG244, TYR191, LEU81, GLU89
**3a**	−5.11	PRO313, TYR191, MET328
**4a**	−6.05	TYR191, GLU89, ARG244, MET328
**5a**	−5.54	PRO326, THR343, TYR191, LEU81, ILE80
**6a**	−5.94	ILE80, GLU89
**7a**	−4.96	PRO313, ILE80, ARG244, TYR191,

**Table 4 molecules-25-03676-t004:** IC_50_ values (μM) of non-prenylated and prenylated tryptophan-containing cyclic dipeptides against HeLa, HepG2, A549 and MCF-7.

IC_50_ (μM)															
HeLa				HepG2				A549				MCF-7			
Substrate		Prenylated		Substrate		Prenylated		Substrate		Prenylated		Substrate		Prenylated	
**1a**	>200	**1b**	95.3	**1a**	>200	**1b**	87.7	**1a**	>200	**1b**	72.4	**1a**	100	**1b**	33.7
**2a**	>200	**2b**	93.1	**2a**	>200	**2b**	>200	**2a**	>200	**2b**	>200	**2a**	>200	**2b**	45.6
**3a**	>200	**3b**	85.2	**3a**	>200	**3b**	>200	**3a**	>200	**3b**	>200	**3a**	>200	**3b**	41.4
**4a**	100	**4b**	75.8	**4a**	>200	**4b**	80.3	**4a**	100	**4b**	61.5	**4a**	100	**4b**	32.3
**5a**	>200	**5b**	94.8	**5a**	>200	**5b**	97.5	**5a**	>200	**5b**	98.0	**5a**	>200	**5b**	42.8
**6a**	>200	**6b**	91.4	**6a**	>200	**6b**	92.8	**6a**	>200	**6b**	99.3	**6a**	>200	**6b**	39.9
**7a**	>200	**7b**	92.2	**7a**	>200	**7b**	82.3	**7a**	>200	**7b**	97.1	**7a**	>200	**7b**	82.5

Values represent mean of three replication.

**Table 5 molecules-25-03676-t005:** Minimal inhibitory concentration (MIC) values of non-prenylated and prenylated tryptophan-containing cyclic dipeptides against Gram-positive bacteria.

MIC (μg·mL^−1^)															
*Bacillus subtilis*				*Staphylococcus aureus*				*Staphylococcus epidermis*				*Staphylococcus simulans*			
Substrate		Prenylated		Substrate		Prenylated		Substrate		Prenylated		Substrate		Prenylated	
**1a**	32	**1b**	16	**1a**	32	**1b**	8	**1a**	128	**1b**	32	**1a**	32	**1b**	4
**2a**	16	**2b**	8	**2a**	8	**2b**	8	**2a**	32	**2b**	16	**2a**	32	**2b**	4
**3a**	256	**3b**	64	**3a**	128	**3b**	64	**3a**	–	**3b**	–	**3a**	128	**3b**	32
**4a**	4	**4b**	0.5	**4a**	8	**4b**	2	**4a**	2	**4b**	1	**4a**	16	**4b**	2
**5a**	32	**5b**	16	**5a**	64	**5b**	16	**5a**	8	**5b**	2	**5a**	32	**5b**	4
**6a**	32	**6b**	16	**6a**	16	**6b**	1	**6a**	–	**6b**	256	**6a**	128	**6b**	16
**7a**	16	**7b**	4	**7a**	16	**7b**	2	**7a**	–	**7b**	256	**7a**	64	**7b**	8
**ampicillin** 64				128				64				–			
**ciprofloxacin** 2				2				2				4			

Values represent mean of three replication, -, no MIC up to 1024 μg·mL^−1.^

**Table 6 molecules-25-03676-t006:** MIC values of non-prenylated and prenylated tryptophan-containing cyclic dipeptides against Gram-negative bacteria.

MIC (μg·mL^−1^)															
*Escherichia coli*				*Klebsiella pneumoniae*				*Proteus mirabilis*				*Pseudomonas aeruginosa*			
Substrate		Prenylated		Substrate		Prenylated		Substrate		Prenylated		Substrate		Prenylated	
**1a**	16	**1b**	2	**1a**	16	**1b**	1	**1a**	32	**1b**	2	**1a**	128	**1b**	16
**2a**	4	**2b**	0.5	**2a**	16	**2b**	2	**2a**	16	**2b**	0.5	**2a**	32	**2b**	4
**3a**	512	**3b**	64	**3a**	–	**3b**	128	**3a**	–	**3b**	256	**3a**	–	**3b**	128
**4a**	8	**4b**	1	**4a**	16	**4b**	4	**4a**	16	**4b**	4	**4a**	16	**4b**	2
**5a**	32	**5b**	2	**5a**	64	**5b**	4	**5a**	64	**5b**	4	**5a**	16	**5b**	1
**6a**	256	**6b**	16	**6a**	–	**6b**	–	**6a**	–	**6b**	–	**6a**	256	**6b**	128
**7a**	512	**7b**	256	**7a**	256	**7b**	16	**7a**	–	**7b**	–	**7a**	128	**7b**	32
**ampicillin** 128				–				64				256			
**ciprofloxacin** 1				2				2				2			

Values represent mean of three replication, -, no MIC up to 1024 μg·mL^−1.^

**Table 7 molecules-25-03676-t007:** MIC values of non-prenylated and prenylated tryptophan-containing cyclic dipeptides against medically important fungi.

MIC (μg·mL^−1^)															
*Aspergillus flavus*				*Candida albicans*				*Cryptococcus gastricus*				*Trichophyton rubrum*			
Substrate		Prenylated		Substrate		Prenylated		Substrate		Prenylated		Substrate		Prenylated	
**1a**	32	**1b**	4	**1a**	16	**1b**	4	**1a**	64	**1b**	8	**1a**	16	**1b**	8
**2a**	32	**2b**	8	**2a**	8	**2b**	1	**2a**	8	**2b**	4	**2a**	16	**2b**	2
**3a**	16	**3b**	2	**3a**	16	**3b**	8	**3a**	64	**3b**	16	**3a**	–	**3b**	256
**4a**	32	**4b**	4	**4a**	16	**4b**	4	**4a**	8	**4b**	1	**4a**	4	**4b**	0.5
**5a**	256	**5b**	64	**5a**	256	**5b**	128	**5a**	–	**5b**	–	**5a**	512	**5b**	64
**6a**	64	**6b**	8	**6a**	128	**6b**	32	**6a**	64	**6b**	4	**6a**	64	**6b**	16
**7a**	32	**7b**	16	**7a**	64	**7b**	4	**7a**	32	**7b**	1	**7a**	64	**7b**	8
**Amphotericin B** 512				16				8				8			

Values represent mean of three replication, -, no MIC up to 1024 μg·mL^−1.^

**Table 8 molecules-25-03676-t008:** MIC values of non-prenylated and prenylated tryptophan-containing cyclic dipeptides against agriculturally important fungi.

MIC (μg·mL^−1^)															
*Fusarium oxysporum*				*Rhizoctonia solani*				*Penicillium expansum*				*Alternaria brassicae*			
Substrate		Prenylated		Substrate		Prenylated		Substrate		Prenylated		Substrate		Prenylated	
**1a**	32	**1b**	16	**1a**	8	**1b**	2	**1a**	16	**1b**	2	**1a**	16	**1b**	4
**2a**	32	**2b**	16	**2a**	4	**2b**	0.5	**2a**	4	**2b**	0.5	**2a**	8	**2b**	1
**3a**	8	**3b**	4	**3a**	32	**3b**	8	**3a**	32	**3b**	2	**3a**	64	**3b**	32
**4a**	2	**4b**	0.5	**4a**	2	**4b**	1	**4a**	4	**4b**	1	**4a**	8	**4b**	2
**5a**	64	**5b**	16	**5a**	64	**5b**	32	**5a**	32	**5b**	16	**5a**	512	**5b**	64
**6a**	64	**6b**	32	**6a**	16	**6b**	4	**6a**	16	**6b**	8	**6a**	–	**6b**	–
**7a**	16	**7b**	2	**7a**	16	**7b**	4	**7a**	8	**7b**	2	**7a**	128	**7b**	32
**Bavistin** 8				16				32				32			

Values represent mean of three replication, -, no MIC up to 1024 μg·mL^−1.^

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
