# Peer review of "C7-Prenylation of Tryptophan-Containing Cyclic Dipeptides by 7-Dimethylallyl Tryptophan Synthase Significantly Increases the Anticancer and Antimicrobial Activities"

_molecules, 2020, doi:10.3390/molecules25163676_

Round 1
Reviewer 1 Report
In this manuscript entitled “A significant increase of antiproliferative, antibacterial and antifungal activity of C7-prenylation in tryptophan-containing cyclic dipeptides by 7-dimethylallyl tryptophan synthase”, there are some problems to be addressed.
- In page 3, line 96, “conversion the lowest rate (11.8 %)”. Yield is different from rate. Are the numbers in % listed in Table 1 product yields after a certain reaction time period or the final yield? Please clarify.
- In lines 108 and 109, “To prove their structures, seven enzyme products (1b-7b) were isolated on HPLC (Figure 1) and subjected to MS and NMR analyses. NMR data and MS data are given as follows.”, please specify the NMR signals which allow authors to determine the prenylation site at position 7, but not at other position. They did mention by comparing other standards, but please be specific, so readers do not need to compare the NMR data.
- In Table 2, please convert Vmax to the numbers with M/s (how many molar concentration of substrate was converted to product, per second) as the unit. It is easier to calculate the kcat/Km. Vmax=kcat[E], so it should be very easy to convert. Per second should be “s-1”, but not “S-1”. DMAPP is the substrate for all the reactions. There is only one number recorded. Does this mean DMAPP has the same Km with respect to different substrates?
- In Figure 3, the H-bonds are not clearly seen. Please use bright color.
- In Discussion, I am interested to learn why C7-prenyl tryptophan-containing cyclic dipeptides have antiproliferative, antibacterial and antifungal activities? What are the mechanisms and the targets of the compounds? Why prenylation at C7 can enhance these activities.
- There are many grammar mistakes needed to be corrected, for example in line 177 “The order of KM, kcat and kcat/KM for seven substrates were inconsistent”, in line 201 “was shown” etc. Even the title is not accurate. I suggest to change to “C7-prenylation of tryptophan-containing cyclic dipeptides by 7-dimethylallyl tryptophan synthase significantly increases the antiproliferative, antibacterial, and antifungal activities”. Please check and correct all through the whole text.
Reviewer 2 Report
In the paper entitled "A significant increase of antiproliferative, antibacterial and antifungal activity of C7-prenylation in tryptophan-containing cyclic dipeptides by 7-dimethylallyl tryptophan synthase" the authors present the C7-enzymatic modification of some Trp-based cyclic dipeptides and the influence of structural change on the biological activity. The idea is interesting, the investigated biological activity profile is complex, but a lot of aspects remain unclear and must be changed:
- The paper needs Major English Language editing, especially for the Abstract and Introduction parts;
- The word "phenylated", used from the title till the end of the Conclusion part, is missused, in my personal opinion; "phenylated" means the introduction of a Phenyl moiety, not the introduction of an other substituent ON a Phenyl moiety; even the enzyme used - DMATS - says that the substituent inserted is in fact dimethyl-allyl (or isopentenyl); this error must be corrected;
- The title says "significant increase of the antiproliferative.....activity", therefore, in order to show that, the authors should also give the IC50 values (Table 4) for substrates 1a-7a, especially since they are already determined (page 8 - line 228);
- Figures 2 and 5 should also be sent as a Supplementary material, individually for each compound, since they are very small and can not be properly analyzed;
- More than 40% of the references are more than 10 years old; the authors should use more "up to date" information;
- I also advice the authors to check the citotoxicity of these compounds on healthy cells (HUVEC for example).
Round 2
Reviewer 2 Report
In this revised version of the manuscript entitled "A significant increase of antiproliferative, antibacterial and antifungal activity of C7-prenylation in tryptophan-containing cyclic dipeptides by 7-dimethylallyl tryptophan synthase" the authors respected most of my suggested requirements. However, there still are some minor chances to do before acceptance:
- minor English language editing; there still are some grammatical errors and some unclear phrases, like "Therefore, we need a more efficient and gentle synthesis strategy, and biotechnology production is the ideal way";
- the authors changed the title to "C7-prenylation of tryptophan-containing cyclic dipeptides by 7-dimethylallyl tryptophan synthase significantly increases the antiproliferative, antibacterial, and antifungal activities" which sound better than the original version, although I would suggest changing the word "antiproliferative" to " anticancer" (since normal cells proliferate too), "antibacterial, and antifungal" to "antimicrobial" (suggested title: "C7-prenylation of tryptophan-containing cyclic dipeptides by 7-dimethylallyl tryptophan synthase significantly increases the anticancer and antimicrobial activities")...the shorter the better.
